# Sub-Domain Aware Granular Segmentation via Fine Tuning Network

## Abstract

Recent advances in deep learning (DL) have led to improved vision-based algorithms. DL-based semantic segmentation, in particular, has enabled precise predictions using Convolutional Neural Networks (CNNs). State-of-the-art CNN-based networks have achieved high accuracy on various datasets in multiple fields, such as building, scene, and object segmentation. However, subdomain shifts between training and test sets within a single domain can cause degraded accuracy in fine-grained segmentation. To counter this, this paper introduces a novel Sub-Domain Adaptation (SDA) framework for fine-grained and granular segmentation, which divides one single domain into multiple sub-domains and optimizes the baseline-network for each sub-domain. The baseline-network is further fine-tuned by recognizing the domain of the input in run-time, leading to more accurate predictions. Benchmarks of scene parsing, autonomous driving, and aerial imagery demonstrate the superior performance of SDA for granular segmentation.[1]

## 1 Introduction

Fine-grained segmentation of objects in an image is an ongoing challenge in a variety of fields, including remote sensing Huang & Gartner (2009), autonomous robotics Lenz et al. (2015), and autonomous driving Sallab et al. (2017). Improvements in accuracy are necessary for practical applications Reed et al. (1994); Zhou et al. (2017). For instance, precise segmentation of buildings in aerial images is vital for creating high-quality digital maps automatically, or for detecting changes in the image for urban planning Kim et al. (2018). Similarly, accurate segmentation of objects in a scene image is critical for autonomous vehicles and robots to respond appropriately to the objects Fridman et al. (2017); Wu et al. (2018).

In recent years, DL methods have proved to be effective for object segmentation in aerial imagery Zhao et al. (2017); Kaiser et al. (2017); Kim et al. (2018); Yue et al. (2019). However, due to the complex shapes, colors, and rotations of objects within cities, such as buildings and roads Lee et al. (2000); Kaiser et al. (2017), achieving accurate segmentation remains challenging. To this end, various architectures have been developed to improve the performance of segmentation. For example, UNetPPL Kim et al. (2018) is a U-Net-based architecture that incorporates multiple pyramid pooling layers Zhao et al. (2017) for extracting multi-scale features. Yue et al. Yue et al. (2019) developed novel layers to further improve the performance of building segmentation. Moreover, Doi and Iwasaki Doi & Iwasaki (2018) applied a focal loss Ross & Dollár (2017) to obtain focused features in aerial images. As aerial images exhibit more vague features, especially boundaries, than other types of vision images, many state-of-the-art models have focused on boundary-oriented segmentation for fine-grained segmentation.

Scene parsing is a popular segmentation task for identifying images into semantic categories such as sky, road, human, and ground Zhou et al. (2017). Despite considerable progress in DL models for semantic segmentation, precise fine-grained segmentation remains a challenge due to the complexity of the different types of datasets. To address this, Romera et al. Romera et al. (2017) proposed a factorized residual layer to improve the efficiency of the DL architecture. BshapeNet Rom Kang & Kim (2018) further improved accuracy by applying bounding shape masks to the Region of Interest. BubbleNet Griffin & Corso (2019), which takes into account a representative frame in the video,

---

[1]Our code is available at https://github.com/Anonymous/Repo

also improved segmentation performance. State-of-the-art (SotA) models have been evaluated using benchmark tests Lin et al. (2014); Zhou et al. (2017).

Recent efforts have demonstrated substantial improvements in segmentation performance (e.g., 50% → 70%; 70% → 85%) using SotA models. In contrast, achieving fine-grained segmentation (e.g., 85% → 90%; 90% → 95%) has proven to be a difficult task. In our preliminary study, we found that a single domain can be further divided into various sub-domains as shown in Fig. 1, and the soft domain shift gaps between different sub-domains lead to a strong deterioration in fine-grained segmentation, as illustrated in Fig. 1. Specifically, the presence of intra-domain gaps between the training and test sets have a particularly detrimental effect on fine-grained segmentation performance. Despite the development of domain adaptation (DA) methods, current approaches are largely designed to reduce domain gaps between two or more different domains, rather than within one single domain. As the domain gaps within a single domain are much smaller than those between two different domains, current DA methods are ineffective in addressing intra-domain gaps.

To address the problem of intra-domain gap, in this work, we introduce the concept of Sub-domain adaptation (SDA) first and then propose an DL framework, dubbed SDA-Net. Instead of the probability-based segmentation, which is known to decrease boundary-oriented segmentation, we use density-based segmentation (Appendix B). To reduce the intra-domain gap, SDA-Net consists of a sub-domain classifier and a baseline-network. The sub-domain classifier identifies sub-domains of inputs and the baseline-network is fine-tuned based on the identified sub-domain via a self-supervised approach. The fine-tuned baseline-network can then provide precise predictions for input images, taking into account the knowledge of intra-domain gaps and sub-domain of inputs.

To summarize, our contributions are below:

- We proposed a self-supervised fine-tuning network (***SDA-Net***) for sub-domain aware granular fine-grained segmentation.
- To achieve fine-grained segmentation, SDA-Net leverages a novel loss function, ***sieve loss***, for self-supervised learning and ***adaptive fine-tuning loss*** for decreasing intra-domain gaps.
- We evaluated our framework on various segmentation benchmarks and demonstrated its superior performance compared with SotA models.

## 2 PRELIMINARY STUDIES

### 2.1 SOFT DOMAIN GAP

A preliminary study of the LoveDA dataset Wang et al. (2021) was carried out to identify sub-domains. Utilizing the t-distributed stochastic neighbor embedding (T-SNE) Van der Maaten & Hinton (2008) algorithm, the dataset was projected into three dimensions and clustered using the non-parametric Density Based Spatial Clustering of Applications with Noise (DBSCAN) Ester et al. (1996). The DBSCAN algorithm split the dataset into three sub-domains, as shown in Fig. 2. Sub-domains were

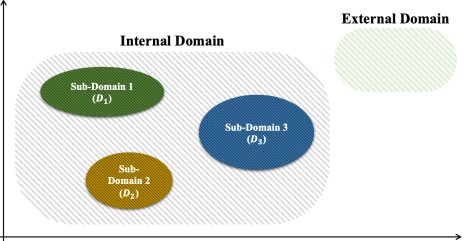

Figure 1: Within a single domain, sub-domains can exist, resulting in an intra-domain gap. Sub-Domain Adaptation (SDA) attempts to reduce this gap, rather than adapting between two distinct domains.

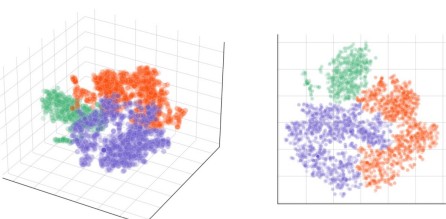

Figure 2: The LoveDA dataset is divided into three sub-domains using the DBSCAN clustering algorithm, and visualized using T-SNE (left: 3D; right: 2D).

split into train and test sets, and a vanilla DL model Ronneberger et al. (2015) was used to train and validate various combinations of sub-domains in a segmentation task. Table 1 shows the results of this study, exploring the effect of soft domain gaps on the performance of the DL model. The results demonstrated that DL performance was maximized (diagonal) when the train sets (columns) and test sets (rows) employed the same sub-domain. Conversely, soft domain gaps lead to a decrease in performance when different sub-domains were used in the train and test sets (e.g., train-set: $D(1)$; test-set: $D(3)$). Furthermore, it is worth noticing that the fully trained DL model ($D(1, 2, 3)$) cannot provide the highest performance when applied to individual sub-domains. This phenomenon, referred to as a "soft domain gap," is analogous to the traditional domain gap and can have a detrimental effect on the DL model's performance. As such, this work aims to reduce the domain gap between sub-domains within a single domain.

Table 1: A preliminary study to investigate the soft domain shift between train and test sets. Results indicate that the highest performance is achieved when the soft domain gap is minimized. Detailed results can be found in Appendix Table 10.

| mIoU | | Train-set | | | |
|---|---|---|---|---|---|
| | | $D(1)$ | $D(2)$ | $D(3)$ | $D(1, 2, 3)$ |
| Test-set | $D(1)$ | **63.93%** | 61.78% | 61.84% | 62.54% |
| | $D(2)$ | 61.64% | **63.75%** | 61.63% | 62.78% |
| | $D(3)$ | 60.63% | 60.39% | **62.58%** | 61.03% |
| | $D(1, 2, 3)$ | 62.69% | 62.49% | 62.46% | **64.67%** |

## 2.2 PROBLEM DEFINITION

In this work, benchmark datasets are divided according to their density (Appendix B). Using this data, $N$ sub-domains are manually grouped under the given dataset ($\mathbb{X}$):

$$D_i^c(\mathbb{X}) = \{\mathcal{X} \mid \frac{i}{N} \leq d^c(\mathcal{X}) < \frac{i+1}{N}, \ \mathcal{X} \in \mathbb{X}\}$$

$$s.t. \ D_i^c(\mathbb{X}) \subset D_i^c \quad \text{and} \quad \bigcup_i^N D_i^c(\mathbb{X}) = D_{\text{all}}^c(\mathbb{X}) = \mathbb{X}. \tag{1}$$

Here, the density of a target object ($c$) is expressed as $d^c(\mathcal{X})$ in an input image, $\mathcal{X} \in \mathbb{R}^{H \times W \times C}$, where $H$ is the height, $W$ is the width, and $C$ is the channel. This density is calculated as the ratio of the number of pixels of $c$-category to the total number of pixels. The set $D_{\text{all}}^i = D_{\text{all}}^i(\mathbb{G})$ represents the domain of mathematically ideal datasets ($\mathbb{G} \supset \mathbb{X}$)[1].

Let $\mathbb{X} \subset \mathbb{R}^{H \times W \times C}$ and $\mathbb{Y} \subset \mathbb{R}^l$ be sets of inputs ($\mathcal{X} \in \mathbb{X}$) and corresponding labels ($\mathcal{Y} \in \mathbb{Y}$) with the number of categories ($l$). Let $\mathcal{C}_\theta$ be a CNN architecture with a set of its parameters ($\theta \in \Theta$), such that $\mathcal{C} : \mathbb{R}^{H \times W \times C} \to \mathbb{R}^{? \times l}$, and the prediction on $\mathbb{X}$ by $\mathcal{C}^\theta$ is provided as $y = \mathcal{C}^\theta(\mathcal{X})$. Then, a cost function ($\mathcal{L} : \Theta \to \mathbb{R}$) to train $\mathcal{C}^\theta$ on $\mathbb{X}$ with a loss function ($L : \mathbb{R}^l, \mathbb{R}^l \to \mathbb{R}$) is defined as below:

$$\mathcal{L}(\mathcal{C}^\theta; \mathbb{X}) = \frac{1}{N} \sum_i^N L(\mathcal{C}^\theta(\mathcal{X}^i), \mathcal{Y}^i), \ \{\mathcal{X}^i \in \mathbb{X}, \mathcal{Y}^i \in \mathbb{Y}\}, \tag{2}$$

where $N$ is the number of samples in $\mathbb{X}$. Hereby, $\theta$ is fully optimized as $\vartheta_{\mathbb{X}}$ via $\vartheta_{\mathbb{X}} = \operatorname{argmin}_\theta \mathcal{L}(\mathcal{C}^\theta; \mathbb{X})$.

Since $\vartheta_{D_{\text{all}}^c} = \operatorname{argmin}_\theta \mathcal{L}(\mathcal{C}^\theta; D_{\text{all}}^c)$ is a fully optimized parameter ideally, $\mathcal{C}^{\vartheta_{D_{\text{all}}^c}}$ can provide precise predictions on all images. However, $\mathcal{C}^{\vartheta_{D_{\text{all}}^c}(\mathbb{X})}$ may provide imprecise predictions since $\mathbb{X}$ is not well-distributed in the real-world. Fig. 3 shows that $\vartheta_{D_{\text{all}}^c}$ reveals the global minima on $\mathcal{L}(\mathcal{C}^\theta; D_{\text{all}}^c)$, while the local-minima on $\mathcal{L}(\mathcal{C}^\theta; D_i^c)$ are caused by the averaging approach in Eq. 2. Consequently, $\mathcal{C}^{\vartheta_{D_{\text{all}}^c}(\mathbb{X})}$ is unable to generate precise predictions on $D_i^c(\mathbb{X})$. In summary, the problem can be defined as follows:

$$\vartheta = \operatorname*{argmin}_\theta \mathcal{L}(\mathcal{C}^\theta; D_{\text{all}}^c) \nRightarrow \forall_\theta \mathcal{L}(\mathcal{C}^\theta; D_i^c) \geq \mathcal{L}(\mathcal{C}^\vartheta; D_i^c). \tag{3}$$

The ensemble DL model ($M^\theta$), consisting of multiple sub-DL models ($M_i^{\theta_i}$) trained for distinct sub-domains ($D_i^c(\mathbb{X})$), can provide accurate predictions on $D_i^c(\mathbb{X})$ as below:

$$\mathcal{X} \in D_i^c(\mathbb{X}) \to M^\theta(\mathcal{X}) = M_i^{\theta_i}(\mathcal{X}). \tag{4}$$

---

[1] $\mathbb{G}$ is an ideal global dataset that contains all possible images in the world. All datasets ($\mathbb{X}$) are subsets of $\mathbb{G}$.

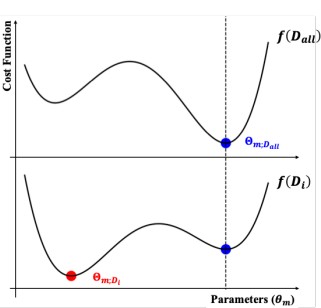

Figure 3: Cost function on $D_{\text{all}}^i$ (top) and $D_c^i$ (bottom)

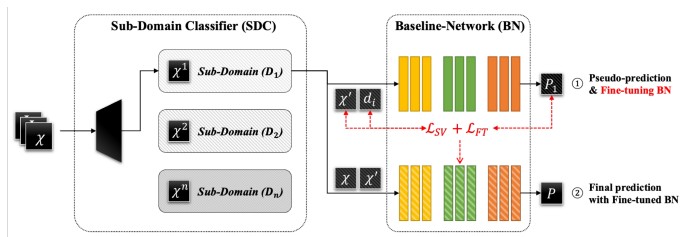

Figure 4: Semantic architecture of the proposed fine-tuning framework. The sub-domain classifier is used to provide a latent feature ($I'$) and predictions on the density ($d_i$). Using this $d_i$, the baseline-network is fine-tuned, enabling it to provide precise predictions using both $I$ and $I'$.

However, the inefficiency of ensemble DL models due to their heavy memory utilization has motivated the development of a novel DL model, dubbed sub-domain adaptation (SDA) via a fine-tuning network (FT-net), that can exhibit the same advantages as the ensemble model, but with improved efficiency. The FT-net pipeline consists of two steps: (1) fine-tuning the parameters of the network to learn the knowledge of a given sub-domain, and (2) utilizing the learned knowledge to provide predictions via SDA. Solving Eq. 3 and realizing SDA, the fine-tuning step provides accurate predictions using Eq. 4, while also achieving memory efficiency. The fine-tuning network provides precise predictions ($y^j$) on the sub-domain ($D_i^c(\mathbb{X})$), as below.

$$1^{\text{st}} : \mathcal{X}^j \in D_i^c(\mathbb{X}) \rightarrow \theta' = \underset{\theta}{\arg\min} \mathcal{L}(\mathcal{C}^\theta; D_i^c(\mathbb{X}) \cup \mathcal{X}^j)$$

$$2^{\text{nd}} : y^j = C^{\theta'}(\mathcal{X}^j) \Rightarrow L(y^j, \mathcal{Y}^j) < L(\mathcal{C}^\theta(\mathcal{X}^j), \mathcal{Y}^j). \tag{5}$$

We propose a novel solution to the problem statement in Eq. 3 through a combination of SDA and FT-net, as shown in Eq. 5.

## 3 METHODS

This section details the architecture of our SDA framework along with its novel loss functions.

### 3.1 OVERALL ARCHITECTURE

Fig. 4 shows the proposed sub-domain adaptation network (SDA-Net) composed of a sub-domain classifier (SDC) and a baseline-network (BN). This framework is capable of addressing the soft domain gap problem by learning to discriminate between sub-domains and mitigating the shift among them. By employing a fine-tuning mechanism, SDA-Net is designed to realize SDA. To this end, our SDA-Net must be aware of the subdomain of the input, which is realized through SDC. The SDC is a CNN ($\mathcal{C}^{\theta_{\text{SDC}}}$) which classifies the input ($\mathcal{X}$) according to its subdomain, producing a probability vector of density ($p^c = (p_1^c, ..., p_i^c); \ p_i^c \in [0,1]; \sum p_i^c = 1$) for each target object ($c \in (1, l)$) in $\mathcal{X}$. Based on $p^c$, the BN is fine-tuned and provides the prediction using $\mathcal{X}$ and $\mathcal{X}'$, which are the latent features extracted by SDC, as inputs. Therefore, the sub-domain of $\mathcal{X}$ is identified as $i = \arg\max \left( \mathcal{C}^{\theta_{\text{SDC}}}(\mathcal{X})|^c \right) = \arg\max_i \left( p_i^c \right)$, and BN is fine-tuned using $\vartheta = \arg\min_{\theta_{\text{BN}}} \mathcal{L}(\mathcal{C}^{\theta_{\text{BN}}}; D_i^c(\mathbb{X}) \cup \mathcal{X})$. The fine-tuned BN then provides the final prediction, $y = \mathcal{C}^\vartheta(\mathcal{X} \oplus \mathcal{X}')$, where $\oplus$ is a Hadamard product.

### 3.2 TRAINING SDA-NET

The training of SDA-Net involves three loss functions: two individual cross-entropy (CE) losses ($L_{\text{CE}}(y, \hat{y}) := \hat{y}\log(y))$) and a novel sieve loss. The network is optimized in a supervised manner, via the CE-loss using $\mathcal{X}^i \in \mathbb{X}$, $\mathcal{Y}^i \in \mathbb{Y}$, and $y = \mathcal{C}^{\theta_{\text{SDA}}}(\mathcal{X}) \in \mathbb{R}^{H \times W \times l}$, such that $L_{\text{CE-S}}(y, \mathcal{Y}) := L_{\text{CE}}(y, \mathcal{Y})$, where $\theta_{\text{SDA}} = \theta_{\text{SDC}} \cup \theta_{\text{BN}}$. Additionally, given $\mathcal{Y} \in \mathbb{Y}$ in the training step, the SDC can

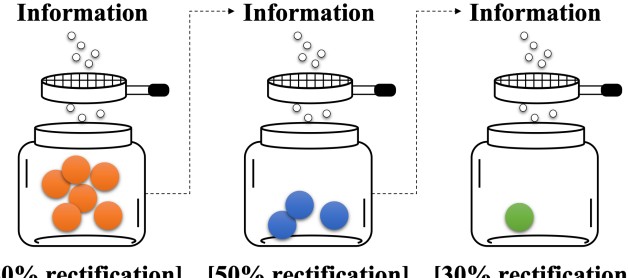

**Information** → **Information** → **Information**

**[80% rectification]** **[50% rectification]** **[30% rectification]**

Figure 5: Information is cumulatively rectified through a series of stages, utilizing ratios of 80%, 50%, and 30%. This leads to a final ratio of 12%, which is obtained by multiplying the three ratios.

be optimized using the calculated density from the input. This is expressed as $L_{\text{CE-D}}\big(y', \bar{d}^c(\mathcal{X}|\mathcal{Y})\big) :=$ $L_{\text{CE}}\big(y', \bar{d}^c(\mathcal{X}|\mathcal{Y})\big)$, where $y' = \mathcal{C}^{\theta_{\text{SDC}}}(\mathcal{X})|^c$, and $\bar{d}^c(\mathcal{X}|\mathcal{Y}) \in \mathbb{R}^l$ is one-hot labeled density.

In this work, we propose a sieve loss, a novel loss for density-based prediction. This loss operates by cumulatively rectifying input information via the activation of ASH ($\text{Æ}$) function Lee et al. (2022). As illustrated in Fig. 5, this process adjusts the input information in a cumulative manner. When predicting densities (i.e., classifying a target object and its area), the resulting activated information should match the density ($d^c(\mathcal{X}|\mathcal{Y})$).

Prior work reported that the $\text{Æ}$ extracts the ratio of the attention area at the activation function level, and its portion is normally distributed as Gaussian distribution. Thus, the activation ratio of each $\text{Æ}$ is calculated from this normal distribution. The final portion of information can then be calculated by multiplying ($\prod$) the attention levels of each $\text{Æ}$. Lastly, the optimization of $\theta_{\text{SDC}}$ is done via the cost function of $\mathcal{L}_{\text{SV}}(\mathcal{C}^{\theta_{\text{SDC}}}; \mathbb{X}, \mathbb{Y})$, which is formulated as follows:

$$\sum_{\mathcal{X}, \mathcal{Y} \in \mathbb{X}, \mathbb{Y}} \left\| \prod_{\in \mathcal{A}(\theta_{\text{SDC}})} \left( \int_{-\infty}^{} \frac{1}{\sqrt{2\pi}} e^{-\frac{x^2}{2}} dx \right) - d^c(\mathcal{X}|\mathcal{Y}) \right\|_2, \tag{6}$$

where $\mathcal{A}(\theta)$ is a set of $\text{Æ}$'s parameters in $\theta$. In summary, SDA-Net is trained using three cost functions, including the CE-loss for segmentation maps ($\mathcal{L}_{\text{CE-S}}$), CE-loss for density classification ($\mathcal{L}_{\text{CE-D}}$), and sieve-loss ($\mathcal{L}_{SV}$). To summarize, SDA-Net is trained with three loss functions as illustrated in Algorithm 1.

---

**Input:** Inputs ($\mathbb{X}$), Labels ($\mathbb{Y}$), SDA-Net ($\theta_{\text{SDA}} = \theta_{\text{SDC}} \cup \theta_{\text{BN}}$)

$epoch = 1$ **to** $EPOCH$

$\nabla\theta_{\text{BN}} \leftarrow \nabla_{\theta_{\text{BN}}} \mathcal{L}_{\text{CE-S}}\Big(\mathcal{C}^{\theta_{\text{SDA}}}; (\mathbb{X}, \mathbb{Y})\Big)$

$\nabla\theta_{\text{SDC}} \leftarrow \nabla_{\theta_{\text{SDC}}} \mathcal{L}_{\text{CE-D}}\Big(\mathcal{C}^{\theta_{\text{SDC}}}; (\mathbb{X}, \bigcup_{\mathcal{X} \in \mathbb{X}, \mathcal{Y} \in \mathbb{Y}} \{\bar{d}^c(\mathcal{X}|\mathcal{Y})\})\Big)$

$\qquad + \nabla_{\theta_{\text{SDC}}} \mathcal{L}_{\text{SV}}\Big(\mathcal{C}^{\theta_{\text{SDC}}}; (\mathbb{X}, \mathbb{Y})\Big)$

**UPDATE** $\theta_{\text{SDA}}$ and $\theta_{\text{BN}}$ with $\nabla_{\theta_{\text{SDA}}}$ and $\nabla_{\theta_{\text{BN}}}$, respectively.

**Algorithm 1:** Training SDA-Net

---

### 3.3 FINE-TUNING SDA-NET

In the fine-tuning step, only BN is fine-tuned using three loss functions: CE-D-loss, sieve-loss, and fine-tuning-loss functions. Let $\mathbb{X}_{\text{tr}} \subset \mathbb{X}$ and $\mathbb{Y}_{\text{tr}} \subset \mathbb{Y}$ be sets of inputs and labels for train-set, and $\mathbb{X}_{\text{te}} \subset \mathbb{X}$ be a set of inputs for test-set, such that $\mathbb{X}_{\text{tr}} \cap \mathbb{X}_{\text{te}} = \emptyset$. Since $\mathbb{Y}_{\text{te}}$ is not provided in the inference step, BN is fine-tuned in an unsupervised manner.

In order to implement Eq. 5, SDC and BN first generate pseudo-predictions ($\mathbf{d}^c = \mathcal{C}^{\theta_{\text{SDC}}}(\mathcal{X})|^c$; $P_1 = \mathcal{C}^{\theta_{\text{SDA}}}(\mathcal{X})$). First, since the target density can be carried out from $P_1$ as $d^c(\mathcal{X}|P_1)$, BN is fine-tuned via $L_1 : \mathcal{L}_{\text{CE-D}}(\mathbf{d}^c, d^c(\mathcal{X}|P_1))$. Second, BN is further optimized to decrease the gap between $\text{Æ}$ activation ratio of $\theta_{\text{BN}}$ and the predicted density by SDC, and thus $\mathcal{L}_2 : \mathcal{L}_{\text{SV}}(\mathcal{C}^{\theta_{\text{BN}}}; \{\mathcal{X}\})$ is carried out as below:

$$\left\| \prod_{\in \mathcal{A}(\theta_{\mathrm{BN}})} \Big( \int_{-\infty}^{\mathcal{X}} \frac{1}{\sqrt{2\pi}} e^{-\frac{x^2}{2}} dx \Big) - \mathcal{C}^{\theta_{\mathrm{SDC}}}(\mathcal{X}) \right\|_2 . \tag{7}$$

Note that only the Æ-related parameters of BN ($\mathcal{A}(\theta_{\mathrm{BN}})$) are fine-tuned using the sieve loss. Hence, the sieve loss enables faster prediction time by decreasing the time to calculate gradients while fine-tuning $\mathcal{A}(\theta_{\mathrm{BN}})$ rather than all parameters of BN ($\theta_{\mathrm{BN}}$), in the inference phase. By adjusting the thresholds, rectifying the target-object-related information of inputs, the sieve loss achieves granular segmentation.

Furthermore, a novel function of fine-tuning-loss (FT-loss) is developed to achieve effective and much faster fine-tuning. As illustrated in Fig. 3, since the globally optimized parameter $\vartheta_{\mathrm{BN};D^c_{\mathrm{all}}(\mathbb{X})}$ is a local minimum on $D^c_i(\mathbb{X})$ ($i = \mathrm{argmax}_i(\mathbf{d}^c)$), further optimization of $\vartheta_{\mathrm{BN};D^c_{\mathrm{all}}(\mathbb{X})}$ on $D^c_i(\mathbb{X})$ keeps the parameters settling in the current local minima.

To address this issue, we introduce a negative term on the gradients ($\nabla_{\theta_{\mathrm{BN}}}\mathcal{L}$) of $\theta_{\mathrm{BN}}$, $\mathcal{L}(\mathcal{C}^{\theta_{\mathrm{BN}}}; D^c_{\mathrm{all}}(\mathbb{X}) - D^c_i(\mathbb{X}))$, which resolves the overfitting problem. This leads to different global minima between $D^c_i(\mathbb{X})$ and $D^c_i(all) - D^c_i(\mathbb{X})$ and thus alternative global minina on $D^c_i(all)$. Consequently, $\mathcal{L}_3 : \mathcal{L}_{\mathrm{FT}}(\mathcal{C}^{\theta_{\mathrm{BN}}}; \mathbb{X})$ can be formulated as:

$$\mathcal{L}_{\mathrm{CE-S}}(\mathcal{C}^{\theta_{\mathrm{BN}}}; D^c_i(\mathbb{X})) - \mathcal{L}_{\mathrm{CE-S}}(\mathcal{C}^{\theta_{\mathrm{BN}}}; D^c_{\mathrm{all}}(\mathbb{X}) - D^c_i(\mathbb{X}).) \tag{8}$$

The use of the negative term in FT-loss has enabled effective and speedy fine-tuning with a relatively small number of epochs. By utilizing a combination of three loss functions, SDA-Net is fine-tuned as illustrated in Algorithm 2. This combination successfully avoids the valley of local minima, thus leading to improved performance.

**Input:** Input ($\mathcal{X}$), SDA-Net ($\theta_{\mathrm{SDA}} = \theta_{\mathrm{SDC}} \cup \theta_{\mathrm{BN}}$)

$epoch = 1$ **to** $k(\leq 10)$

**Pseudo-prediction:** $\mathbf{d}^c = \mathcal{C}^{\theta_{\mathrm{SDC}}}(\mathcal{X})|^c$, $P_1 = \mathcal{C}^{\theta_{\mathrm{SDA}}}(\mathcal{X})$

$\nabla \theta_{\mathrm{BN}} \leftarrow \nabla_{\theta_{\mathrm{BN}}} \mathcal{L}_{\mathrm{CE-D}}\Big( d^c(\mathcal{X}|\mathcal{C}^{\theta_{\mathrm{SDC}}}); (\{\mathcal{X}\}, \{\mathbf{d}^c\})\Big)$

$\qquad + \nabla_{\theta_{\mathrm{SDC}}} \mathcal{L}_{\mathrm{SV}}\Big( \mathcal{C}^{\theta_{\mathrm{BN}}}; \{\mathcal{X}\}\Big)$

$\qquad + \nabla_{\theta_{\mathrm{SDC}}} \mathcal{L}_{\mathrm{FT}}\Big( \mathcal{C}^{\theta_{\mathrm{BN}}}; \mathbb{X}\Big)$

**UPDATE** $\theta_{\mathrm{BN}}$ with $\nabla_{\theta_{\mathrm{BN}}}$.

**Final prediction:** $P = \mathcal{C}^{\theta_{\mathrm{SDC}} \cup \theta_{\mathrm{BN}}}(\mathcal{X})$

**Output:** $P$

**Algorithm 2:** Fine-tuning SDA-Net

## 4 EXPERIMENTS

### 4.1 DESCRIPTION OF EXPERIMENTAL SET UP

To evaluate our framework, we employed two DL models: U-Net Ronneberger et al. (2015) and CCNet Huang et al. (2019). U-Net is a popular basic model for segmentation, while CCNet contains an attention module, making it an advanced model relative to U-Net. CCNet was used as the Baseline-Network for SDA-Net. Additionally, InternImage (II) Wang et al. (2022), which is a state-of-the-art model for the segmentation of scene parsing benchmarks, and LoveDA (LDA) Wang et al. (2021), a state-of-the-art model for the multi-categorical segmentation of remote-sensing benchmarks, were employed for comparison. Furthermore, Segmenter (ST) Strudel et al. (2021), SiamixFormer (SF) Mohammadian & Ghaderi (2022) and Hybrid-ASPP (H-ASPP) Luo et al. (2022) were employed as the comparison transformer models for vanilla, aerial imagery, and autonomous driving, respectively.

### 4.2 MODEL IMPLEMENTATIONS

We implemented our framework based on the ResNet-18 He et al. (2016) for SDC and CCNet for BN. We replaced all activation functions of ResNet-16 and CCNet with the Æ activation function to

Table 2: Detailed description of the datasets. To validate, 10-fold cross-validation was employed for each dataset.

| Dataset | # of Images | # of Train | # of Test | # of Labels |
|---|---|---|---|---|
| WHU | 816 | 735 | 81 | 2 |
| LoveDA | 4191 | 3,772 | 419 | 8 |
| BDD100K | 8,000 | 7,200 | 800 | 20 |
| GTA5 | 24,966 | 22,470 | 2,496 | 27 |
| ADE20K | 27,574 | 24,817 | 2,757 | 150 |

implement the sieve loss, and employed DeepLabV3 Chen et al. (2017) as the baseline-network for the CCNet. Our framework and other comparison DL models were implemented and evaluated on four A5000 GPUs, Xeon(R) Gold CPUs, and 512GB Memory, using Python 3.10, TensorFlow 2, and PyTorch in an Ubuntu 20.04 environment. For a fair comparison, all DL models were trained with a batch size of 10, and images were resized to $256 \times 256$ Bottou (2010). As the WHU dataset was insufficient to provide a large enough number of images, one image in the dataset was cropped into four images of size $256 \times 256$. Moreover, the ADAM optimizer was used with the default parameters Kingma & Ba (2014), and all models were initialized based on a Gaussian distribution with mean and standard deviation values of 0 and 1, respectively.

## 4.3 ABLATION STUDY

In order to assess the performance of different models, we carried out ablation studies with a variety of baseline models for SDC and BN, as well as varying the number of domains. Additionally, we tested the efficacy of our novel loss functions, the sieve and fine-tuning losses.

In Table 3, the performance of different DL models is evaluated in terms of mIoU, the number of parameters, and Floating-point arithmetic (FLOPs). SDA-Net with CCNet and ResNet-18 as the baseline models yielded the best results, compared with other combinations. Experiments show that the number of parameters of the model is not necessarily indicative of the performance, as a larger number of parameters (ResNet-18 vs ResNet-152) does not guarantee a better performance in a rather straightforward density prediction task. In contrast, U-Net, which has a smaller number of parameters compared with CCNet, yielded lower mIoU values, due to the difficulty of the segmentation task. Additionally, Fig. 6 shows that SDA-Net with CCNet exhibits more accurate predictive performance on the LoveDA dataset. Thus, ResNet-18 and CC-Net were selected as the baseline networks for the optimal SDA-Net.

Fig. 6 and Table 4 show that, when predicting the LoveDA dataset, SDA-Net with a range of different numbers of sub-domains yields varying levels of mIoU. Experimental results on the ablation study confirm that augmenting the number of sub-domains based on density leads to an improvement in SDA-Net's performance. However, the distinction between Ours-10 and Ours-20 is minor. This implies that further increases in the number of sub-domains produce only a slight enhancement in performance, while also requiring a substantial number of parameters. It was observed that Ours-10 has a more uneven distribution than Ours-3 and Ours-5, due to its density-wise predictions.

Fig. 7 illustrates the predictions of CCNet and SDA-Net on each category of the LoveDA dataset. CCNet provides imbalanced distributions of predictions depending on the number of samples in

Table 3: Ablation study on the baseline network for SDC and BN. FLOPs are calculated on an input with a size of $256 \times 256 \times 3$, and the best performance values are highlighted in **bold**.

| SDC | BN | mIoU | # params | FLOPs |
|---|---|---|---|---|
| ResNet-18 | U-Net | 61.88 (12.8) | **44.3** | **22.6** |
| | CCNet | **65.79 (15.7)** | 84.5 | 98.6 |
| VGG19 | U-Net | 61.08 (13.9) | 176.3 | 50.5 |
| | CCNet | 62.60 (13.6) | 216.5 | 126.6 |
| ResNet-152 | U-Net | 64.90 (16.1) | 92.8 | 37.8 |
| | CCNet | 64.88 (15.7) | 133.0 | 113.9 |

Table 4: Ablation study on the different numbers of sub-domains for SDA-Net. Ours-$k$ indicates $k$ numbers of sub-domains with SDA-Net.

| mIoU | LoveDA | WHU | BDD100K | GTA5 | ADE20K |
|---|---|---|---|---|---|
| Ours-3 | 50.1 | 86.8 | 49.8 | 72.2 | 59.4 |
| Ours-5 | 50.7 | 87.2 | 50.0 | 72.6 | 59.8 |
| Ours-10 | 52.4 | 88.7 | 52.0 | 74.4 | 61.5 |
| Ours-20 | 52.5 | 88.8 | 52.0 | 74.6 | 61.7 |

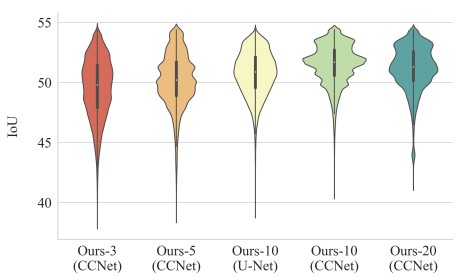 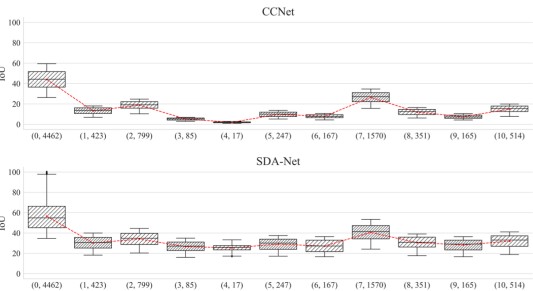

Figure 6: Violin chart for diverse versions of SDA-Net

Figure 7: A comparison of mIoU values for different categories by the CCNet (top) and SDA-Net (bottom). The x-axis indicates the category index and the number of samples.

each category. However, SDA-Net, which can provide predictions based on the sub-domains of each category, offers balanced predictions as well as higher mIoU values than CCNet. This performance on granular segmentation of SDA-Net is demonstrated in Fig. 7.

Table 5: Ablations study on loss functions of sieve loss ($\mathcal{L}_{SV}$) and fine-tuning loss ($\mathcal{L}_{FT}$). '−' symbol indicates 'without'.

| | LoveDA | WHU | BDD100K | GTA5 | ADE20K |
|---|---|---|---|---|---|
| Ours | 52.4 | 88.7 | 52.0 | 74.4 | 61.5 |
| Ours - $\mathcal{L}_{SV}$ | 48.5 | 82.5 | 46.5 | 66.7 | 50.0 |
| Ours - $\mathcal{L}_{FT}$ | 48.0 | 84.0 | 47.7 | 66.2 | 57.5 |
| Ours - $\mathcal{L}_{SV}$ - $\mathcal{L}_{FT}$ | 47.3 | 72.4 | 43.7 | 65.1 | 49.3 |

In Table. 5, different versions of SDA-Net with or without the sieve and fine-tuning losses were evaluated. Large gaps between the SDA-Net without one loss function and the SDA-Net without two loss functions demonstrate the effectiveness of the sieve and fine-tuning loss functions in improving the performance of SDA-Net. Furthermore, the simultaneous use of both loss functions promotes the sub-domain optimization of BN, resulting in the highest mIoU values on all datasets for the SDA-Net with the sieve and fine-tuning loss functions.

## 4.4 COMPARISON ANALYSIS

Table 6: Evaluations on the prediction time, in terms of the number of parameters, FLOPs, and prediction time. FLOPs are measured using an input size of $896 \times 896 \times 3$.

| | sice | # Params | # FLOPs | Time (ms) |
|---|---|---|---|---|
| CCNet | 896 | 71.3M | 0.94B | 73 |
| H-ASPP | 896 | - | 0.62B | 56 |
| II | 896 | 1.31B | 4.64B | 107 |
| Ours | 896 | 90.7M | 1.04B | 116 |

We compared SDA-Net to other DL models in terms of the number of parameters and prediction time as shown in Table 6. SDA-Net demonstrated a comparable number of parameters and prediction time to the other DL models, despite its fine-tuning process. For instance, SDA-Net exhibited only a 0.1B increase in FLOPs, despite having a larger number of parameters than CCNet. Furthermore, SDA-Net contained fewer parameters than InternImage (II), resulting in fewer calculations required for prediction time. Thus, SDA-Net achieved a similar prediction time to II in spite of its fine-tuning process. This is because the fine-tuning is applied to a small number of parameters over a limited number of epochs ($\leq 10$), allowing for faster prediction.

As shown in Table 7, in addition, the proposed SDA-Net yielded more outstanding performances than other compared DL models, including vision transformers and SotA models, for scene parsing and

Table 7: Comparison of SDA-Net with other compatible DL models on four datasets: WHU, BDD100K, GTA5, and ADE20K. Results were measured in terms of mIoU and standard deviation values.

|  | WHU | BDD100K | GTA5 | ADE20K |
|---|---|---|---|---|
| UNet | 68.7 ($\pm$3.53) | 43.4 ($\pm$1.29) | 65.0 ($\pm$1.53) | 49.1 ($\pm$2.30) |
| CCNet | 70.7 ($\pm$4.15) | 43.6 ($\pm$1.49) | 65.1 ($\pm$1.99) | 50.0 ($\pm$1.91) |
| ST | 74.6 ($\pm$3.72) | 44.5 ($\pm$1.74) | 66.2 ($\pm$2.12) | 52.2 ($\pm$2.88) |
| LDA | 82.0 ($\pm$4.39) | 47.1 ($\pm$1.68) | 66.7 ($\pm$1.55) | 53.9 ($\pm$2.25) |
| II | 80.2 ($\pm$4.55) | 48.4 ($\pm$1.67) | 67.0 ($\pm$2.05) | 55.4 ($\pm$2.86) |
| SEPC | 79.3 ($\pm$3.04) | 48.0 ($\pm$1.82) | 68.8 ($\pm$2.02) | 55.6 ($\pm$2.17) |
| SF | 78.4 ($\pm$4.56) | 45.8 ($\pm$2.12) | 67.2 ($\pm$2.39) | 55.1 ($\pm$2.99) |
| H-ASPP | 79.4 ($\pm$3.92) | 44.7 ($\pm$1.33) | 69.3 ($\pm$1.61) | 56.7 ($\pm$2.49) |
| Ours-3 | 86.8 ($\pm$3.82) | 49.8 ($\pm$2.10) | 72.2 ($\pm$1.83) | 59.4 ($\pm$2.20) |
| Ours-5 | 87.2 ($\pm$4.87) | 50.0 ($\pm$2.11) | 72.6 ($\pm$1.53) | 59.8 ($\pm$1.87) |
| Ours-10 | 88.7 ($\pm$4.34) | 52.0 ($\pm$1.78) | 74.4 ($\pm$1.84) | 61.5 ($\pm$2.47) |

remote-sensing images. Specifically, compared with transformers and SotA DL models, SDA-Net with ten sub-domains showed +6.7%, 3.6%, 5.1%, and 4.8% improvement in WHU, BDD100K, GTA5, and ADE20K datasets, respectively. Additionally, SDA-Net yielded an average of +12.6% improvement when compared with vanilla models of U-Net and CCNet (see Table 7).

Tables 8 and 9 and Fig. 8 show the qualitative results of the SDA-Net compared with other compared DL models. In Tables 8 and 9, the detailed analysis shows that the SDA-Net provides more balanced mIoU values in comparison to other compared DL models, regardless of the varying distributions of target object densities. This demonstrates the effectiveness of our SDA-Net in density-based predictions, thereby bridging the soft-domain gaps. In Fig. 8, the portion of each target object varies greatly between categories, which has caused the vanilla model and other SotA models to produce mis-predicted segmentation maps with imprecise boundaries. The SDA-Net, on the other hand, was able to provide finer segmentation masks, indicating that the sub-domain adaptation of the SDA-Net allows for granular segmentation.

Table 8: A comparison analysis of the mIoU values of DL models on the LoveDA dataset.

|  | None | building | road | water | barren | forest | agriculture | bg | mIoU |
|---|---|---|---|---|---|---|---|---|---|
| UNet | 48.1 | 47.2 | 47.5 | 46.6 | 46.9 | 47.0 | 47.1 | 46.9 | 47.2 |
| CCNet | 49.3 | 47.7 | 48.2 | 46.6 | 47.1 | 47.2 | 47.5 | 47.2 | 47.6 |
| ST | 50.0 | 48.7 | 49.2 | 47.9 | 48.2 | 48.4 | 48.6 | 48.3 | 48.7 |
| LDA | 50.6 | 49.6 | 50.0 | 49.0 | 49.2 | 49.3 | 49.5 | 49.3 | 49.6 |
| II | 51.1 | 49.8 | 50.3 | 49.0 | 49.3 | 49.4 | 49.7 | 49.4 | 49.7 |
| SEPC | 50.5 | 49.8 | 50.1 | 49.4 | 49.6 | 49.6 | 49.8 | 49.6 | 49.8 |
| SF | 50.9 | 49.5 | 50.0 | 48.7 | 49.0 | 49.2 | 49.4 | 49.1 | 49.5 |
| H-ASPP | 49.8 | 49.2 | 49.4 | 48.8 | 49.0 | 49.0 | 49.1 | 49.0 | 49.2 |
| Ours-3 | 51.3 | 50.3 | 50.6 | 49.7 | 49.9 | 50.0 | 50.2 | 50.0 | 50.2 |
| Ours-5 | 51.8 | 50.8 | 51.2 | 50.2 | 50.4 | 50.6 | 50.7 | 50.5 | 50.8 |
| Ours-10 | 52.6 | 52.4 | 52.5 | 52.2 | 52.3 | 52.2 | 52.4 | 52.4 | 52.4 |

## 5 CONCLUSION

This paper has introduced a novel framework, SDA-Net, which is capable of addressing the intra-domain and soft-domain gaps in granular segmentation. The SDA-Net recognized the index of the sub-domain of inputs and fine-tunes the baseline-network of SDA-Net, allowing precise predictions on the inputs. In order to achieve this, two novel loss functions, sieve and fine-tuning losses, were proposed. The sieve loss provided attention-based gradients with a small computational cost, while the fine-tuning loss provided negative terms to escape from the local minimum caused by the soft-domain gaps. Experimental results demonstrated that the SDA-Net significantly enhanced the segmentation performance. This novel framework can employ any other advanced SotA models for more enhanced segmentation without complex implementations, although finding the best SotA models and hyperparameters are required for further enhancement in segmentation tasks, which is subject to our future work.

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

## A  EXPERIMENTAL RESULTS

Table 9: A comparison analysis of mIoU values of different DL models on the BDD100K dataset.

| | bg | road | sidewalk | building | wall | fence | pole | traffic light | traffic sign | vegetation | terrain | sky | person | rider | car | truck | bus | train | motorcycle | bicycle |
|---|---|---|---|---|---|---|---|---|---|---|---|---|---|---|---|---|---|---|---|---|
| UNet | 43.0 | 46.0 | 45.4 | 44.7 | 44.6 | 41.0 | 45.3 | 44.5 | 43.2 | 44.2 | 44.3 | 43.0 | 45.6 | 44.2 | 45.9 | 45.5 | 44.0 | 45.0 | 44.7 | 43.4 |
| CCNet | 43.1 | 46.3 | 45.7 | 44.9 | 44.7 | 40.9 | 45.5 | 44.7 | 43.3 | 44.3 | 44.5 | 43.1 | 45.9 | 44.4 | 46.3 | 45.8 | 44.1 | 45.3 | 44.9 | 43.4 |
| Vit | 44.2 | 46.7 | 46.2 | 45.6 | 45.5 | 42.5 | 46.1 | 45.5 | 44.3 | 45.2 | 45.3 | 44.2 | 46.4 | 45.2 | 46.7 | 46.3 | 45.0 | 45.9 | 45.6 | 44.5 |
| LDA | 45.3 | 47.8 | 47.4 | 46.8 | 46.6 | 43.6 | 47.2 | 46.6 | 45.5 | 46.3 | 46.4 | 45.3 | 47.5 | 46.3 | 47.8 | 47.5 | 46.1 | 47.1 | 46.7 | 45.6 |
| II | 46.1 | 49.4 | 48.7 | 47.9 | 47.8 | 43.9 | 48.6 | 47.8 | 46.3 | 47.4 | 47.5 | 46.1 | 49.0 | 47.4 | 49.3 | 48.9 | 47.1 | 48.3 | 47.9 | 46.5 |
| SEPC | 46.4 | 48.1 | 47.8 | 47.3 | 47.3 | 45.2 | 47.7 | 47.2 | 46.5 | 47.0 | 47.1 | 46.4 | 47.9 | 47.0 | 48.1 | 47.8 | 46.9 | 47.6 | 47.3 | 46.6 |
| SF | 44.4 | 47.5 | 46.9 | 46.1 | 46.0 | 42.3 | 46.8 | 46.0 | 44.6 | 45.6 | 45.8 | 44.4 | 47.1 | 45.6 | 47.5 | 47.0 | 45.4 | 46.5 | 46.1 | 44.7 |
| H-ASPP | 45.5 | 48.5 | 47.9 | 47.2 | 47.1 | 43.5 | 47.8 | 47.0 | 45.7 | 46.7 | 46.8 | 45.5 | 48.1 | 46.7 | 48.5 | 48.0 | 46.5 | 47.6 | 47.2 | 45.8 |
| Ours-5 | 48.4 | 51.1 | 50.5 | 49.9 | 49.8 | 46.7 | 50.4 | 49.8 | 48.6 | 49.5 | 49.6 | 48.4 | 50.8 | 49.5 | 51.0 | 50.7 | 49.3 | 50.2 | 49.9 | 48.7 |
| Ours-10 | 50.1 | 52.4 | 51.9 | 51.3 | 51.3 | 48.5 | 51.7 | 51.2 | 50.1 | 50.9 | 51.0 | 50.2 | 52.1 | 50.9 | 52.3 | 52.0 | 50.8 | 51.6 | 51.3 | 50.3 |

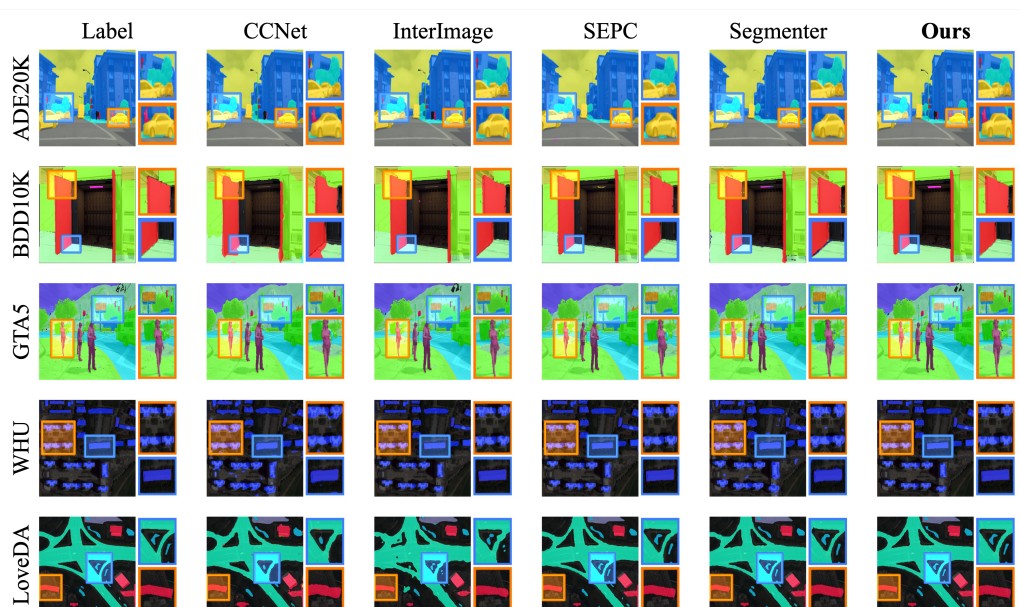

Figure 8: Representative predicted segmentation results on five individual datasets, using vanilla (CCNet), SotA for scene parsing (InternImage), SotA for remote sensing (SEPC), vision transformer (Segmenter), and SDA-Net (Ours).

## B  DENSITY-BASED SEGMENTATION

### B.1  PROBABILITY-BASED SEGMENTATION

In the early era, deep learning (DL)-based segmentation algorithms have been developed from Fully Convolutional Networks Long et al. (2015), producing simple yet effective networks Noh et al. (2015); Ronneberger et al. (2015); Quan et al. (2016); Badrinarayanan et al. (2017). More recently, advanced deep learning models have exhibited state-of-the-art performance in many applications Tao et al. (2020); Zoph et al. (2020); Huang et al. (2021); Bao et al. (2021); Niu et al. (2021); Gupta et al. (2021), with some models designed specifically for certain applications Lin et al. (2017); Zhuang (2018); Bai & Zhou (2020); Byra et al. (2020). However, it has been found that conventional segmentation models can exhibit degraded accuracy when the size of the target object (e.g. buildings, roads, cars, pedestrians, etc.) is different between training and test sets Han & Davis (2011); Zhang et al. (2017; 2019). To address this, we introduce the concept of density—the number of pixels per total number of pixels—and how it correlates with the probability of a segmentation output in $\mathbb{R}^{H \times W \times C}$. Segmentation masks are then generated from the resulting probability distribution and a threshold, usually set at 0.5.

**Definition I.** Let $p_c(h, w; I^{(i)})$ be the random variable to be classified as a target object ($c$) at the pixel of $(h, w)$ in the image ($I^{(i)}$), and $P_c(I)$ be a probability distribution of the set of images ($I^{(i)} \in I$). Then, $0 \leq p_c(h, w; I^{(i)}) \leq 1$, $\sum_c p_c(h, w; I^{(i)}) = 1$, and $p_c(h, w; I^{(i)}) \sim P_c(I)$.

**Definition II.** Let $\Omega_c(h, w; I)$ be a category ($c$) recognition function at pixel $(h, w)$ in $I^{(i)}$. Then, $\Omega_c(h, w; I^{(i)})$ is 1 *iff* $\underset{x}{\mathrm{argmax}}\, p_x(h, w; I^{(i)})$ = c, otherwise 0.

**Definition III.** Let $d_c : I^{(i)} \to \mathbb{R}$ be the site area function of the target object ($c$) in image ($I^{(i)}$). Then, $d_c(I^{(i)}) = \frac{1}{HW} \sum_h^H \sum_w^W \Omega(h, w; I)$ with the image of height ($H$) and width ($W$). In addition, let $D_c(I)$ be a site area distribution of the set of images ($I^{(i)} \in I$). Then, $d_c(I^{(i)}) \sim D_c(I)$.

**Definition IV.** Let $O(S, G; M)$ be the optimization of a DL model (M) using two probability distributions of $S$ and $G$. Then, $M$ is optimized by approximating $S$ to $G$.

Generally, $p_c(h, w; I^{(i)})$ is denoted as a softmax output, and $S$ and $G$ are the predicted segmentation maps and the corresponding ground truths, respectively. The conventional optimization of $M$ is known as the probability-based segmentation, which aims to approximate the output probability ($p_c(h, w; S) \sim P_c(S)$) to the ground truth ($p_c(h, w; G) \sim P_c(G)$); $O(P_c(S), P_c(G); M)$. In contrast, site area-based segmentation approximates the site area of target objects in the predicted segmentation map ($d_c(S) \sim D_c(S)$) to the ground truth ($d_c(G) \sim D_c(G)$); $O(D_c(S), D_c(G); M)$.

**Theorem I.** Let $A$ and $A'$ be the training set and test set, respectively. Then, $O(P_c(S), P_c(G); M) \implies O(P_c(S'), P_c(G'); M)$, but $O(P_c(S), P_c(G); M) O(D_c(S'), D_c(G'); M)$.

**Theorem II.** Let $A$ and $A'$ be the training set and test set, respectively. $O(P_c(S), P_c(G); M) \wedge d_c(S) \sim D_c(S) d_c(S') \sim D_c(S')$.

Theorem I and II indicate that the segmentation performance decreases when the site area differs between the training and test sets when using probability-based segmentation as opposed to site area-based segmentation.

## B.2 DENSITY-BASED SEGMENTATION

Density-based segmentation algorithms have been widely studied in recent years. Zhang *et al*. Zhang et al. (2017) proposed a method that applied density-based clustering and nodule segmentation to localize lung nodules in CT sequence images. Han *et al*. Han & Davis (2011) utilized a density function for a multiple feature integration machine learning (ML) algorithm, which was applied to a classification task. Zhang *et al*. Zhang et al. (2019) developed a density-based unsupervised segmentation technique that incorporated density-based clustering and sensitive parameter setting techniques. Despite these advances, the performance of density-based segmentation has remained limited due to the use of ML- or clustering-based methods without the aid of CNNs.

While improving the segmentation performance of a DL model, we found that the predicted probability distribution of target objects is dependent on a training set. Specifically, when training a DL model with a target object density of 10-20%, the model's predicted performance is highest when the corresponding test set has the same range of density. However, if the test set's object density is outside the training set's range, such as 80-90%, the model's performance is degraded.

A cross-entropy-based optimization derived from KL Divergence is used to train a network to predict a probability distribution akin to the source input. This technique was demonstrated by the successful prediction of probability distributions of the test images, which were comparable to those of the training images. Let $p^{(i)}(h, w)$ be the random variable to be classified as the target object and let $\Psi$ be the function that accepts the input image ($I^{(i)}$) and pixel locations of $(h, w)$, and generates $p^{(i)}(h, w)$ as follows:

$$\Psi((h, w); I^{(i)}) = p^{(i)}(h, w). \tag{9}$$

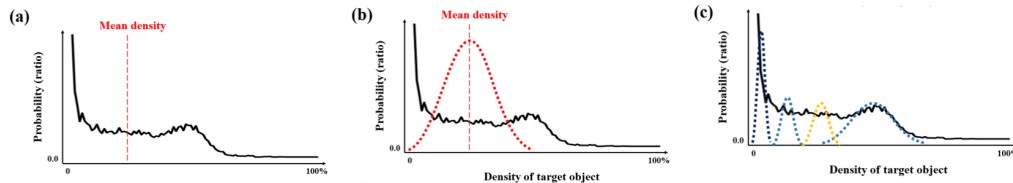

Figure 9: (a) Density distribution of the dataset, (b) the mean value of the density and the approximated Gaussian distribution of density distribution in (a), and (c) the proposed density-based segmentation method. Multiple deep learning models are used for each density.

In the classification task, a probability of $p^{(i)}(h, w)$ greater than 0.5 is used as the threshold to predict the target object. The calculated density $d_c$ of the target object can then be determined:

$$d_c^{(i)} = \frac{1}{HW} \sum_h^H \sum_w^W S(p^{(i)}(h, w) - 0.5), \tag{10}$$

where $S$ represents the Heaviside Step Function, and H and W indicate the height and width of $I^{(i)}$, respectively. In contrast, we can define the real densities of targets in $I^{(i)}$ as $d_r$, as follows:

$$d_r^{(i)} = \frac{1}{HW} \sum_h^H \sum_w^W D(h, w), \tag{11}$$

where $D(h, w)$ is 1 if the pixel in $I^{(i)}$ at $(h, w)$ location is classified as the target object, and otherwise $D(h, w)$ is 0. Therefore, two random variables of $p^{(i)}(h, w)$ and $D(h, w)$ are in the closed range of $[0, 1]$ and the set of $\{0, 1\}$. Here, the cross-entropy is used as the objective function to optimize DL models as follows:

$$\mathcal{L}_\rangle = G^{(i)} \log \left( \frac{G^{(i)}}{P^{(i)}} \right), \tag{12}$$

where $G$ and $P$ are ground truth and the prediction by the DL model. In this paper, the objective function with $p^{(i)}(h, w)$ and $D(h, w)$ is defined as follows:

$$\mathcal{L} = \sum_i^N D(h, w) \log \left( \frac{D(h, w)}{p^{(i)}(h, w)} \right), \tag{13}$$

where $N$ is the total number of images in the training set. However, to achieve a similar density distribution between $d_r$, which is determined by annotations, and $d_c$, which is determined by a DL model, the following must be met:

$$d_r^{(i)} \log \left( \frac{d_r^{(i)}}{d_p^{(i)}} \right) = d_r^{(i)} \log \left( \frac{\frac{1}{HW} \sum_h^H \sum_w^W D(h, w)}{\frac{1}{HW} \sum_h^H \sum_w^W H(p^{(i)}(h, w) - 0.5)} \right). \tag{14}$$

However, the optimization of Eq. 13 does not guarantee Eq. 14 due to the information loss caused by the rectification by the Heaviside Step Function in Eq. 14. This leads to different expectations $(E)$ for $d_c$ and $d_r$ due to the same reasons, as follows:

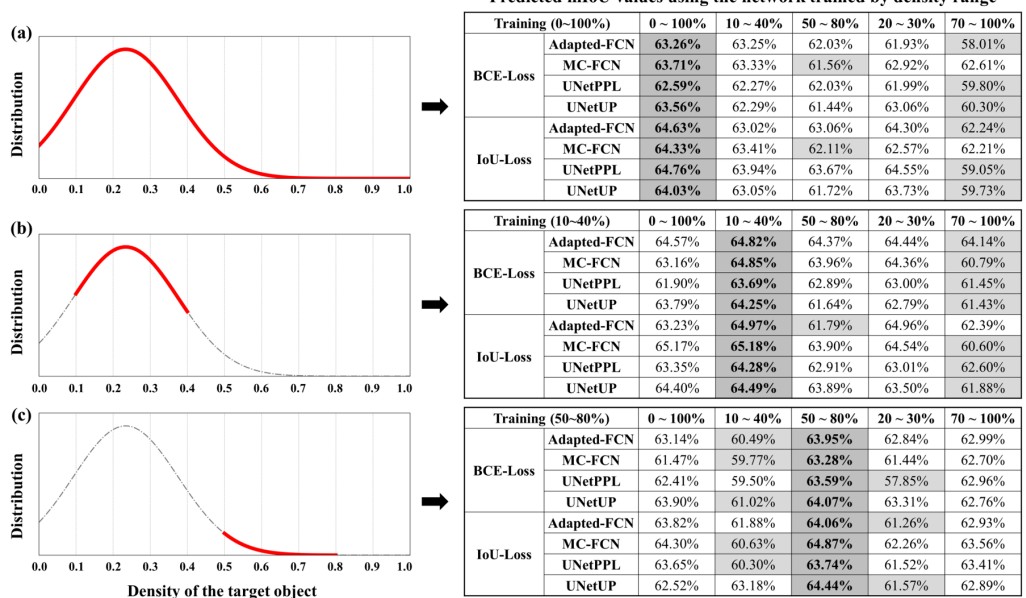

Figure 10: The graphs of the density distribution of the dataset and the tables of the predicted mIoU values by each network and loss function. Red-colored lines are used to indicate images used for training the networks. The tables show the predicted mIoU values for images in specific density ranges, with the highest mIoU values highlighted in **darkgray** and the lowest IoU values highlighted in **gray**. (a) All images in the dataset are used for training the network. (b) Images in the density range of [0.1, 0.4] are used for training the network. (c) Images in the density range of [0.5, 0.8] are used for training the network. The tables demonstrate that the bigger the difference between the densities of the training set and test set, the poorer the performance.

$$
\begin{aligned}
E(d_c^{(i)}) &= \frac{1}{HW} \sum_h^H \sum_w^W E(H(p^{(i)}(h, w) - 0.5)) \\
E(d_r^{(i)}) &= \frac{1}{HW} \sum_h^H \sum_w^W E(D(h, w)).
\end{aligned}
\tag{15}
$$

If there is an optimal function that can accurately predict target objects from input images, it is expected that the value of $E(d_c^{(i)})$ and $E(d_r^{(i)})$ should be the same. However, due to the fact that the training set cannot cover all real images in the world, and that to maintain memory efficiency, the batch images are utilized with SGD, there is a likely gap between $E(d_c^{(i)})$ and $E(d_r^{(i)})$. Therefore, if $d_c$ is fully optimized to $d_{r_1}$ in one subset of training-set ($T_1$), then $d_c$ differs from $d_{r_2}$ in another subset of training or test set ($T_2$) as the following:

$$
\exists \epsilon > 0 \lim_{x \to 0} |E(d_c) - E(d_{r_1})| < x \Rightarrow |E(d_c) - E(d_{r_2})| > \epsilon.
\tag{16}
$$

Therefore, the density distribution of the predictions of the DL model, which is optimized for $T_1$, differs from the density distribution of another set, such as the test set, $T_2$. To illustrate, the density distribution of the dataset is shown in Appendix Fig. 9(a), and the density distribution is approximated as a Gaussian distribution in Appendix Fig. 9(b) to calculate the KL Divergence. In this case, the DL model is optimized to Appendix Fig. 9(a), resulting in a density distribution of the predictions that more closely resembled that of Appendix Fig. 9(b). This demonstrated a dependence of the density of the predictions on that of the source input images. To address this issue, a density-based segmentation method was developed. This method allowed the DL model to recognize the densities

of the targets, and to segment the input images using different sub-models for suitable density ranges, as illustrated in Appendix Fig. 9(c).

Fig. 10 in the Appendix demonstrates the correlation between the densities of the train set and the test set, and the prediction accuracy. The highest performance, as measured by IoU, is attained when the densities of the train set and the test set are the same. Conversely, when the densities of the train set and the test set are significantly different, the prediction accuracies are significantly decreased. To predict a test set by a network that is optimized using a train set with similar density, we proposed a method to calculate the density of an object and segment objects with a network trained for the same density range. Since it is not possible to accurately calculate the density prior to segmenting the target object, we designed a CNN-based structure capable of predicting the density of the object.

## C  DETAILS OF PRELIMINARY STUDY

The LoveDA dataset was divided into three sub-domains, $D(1)$, $D(2)$, and $D(3)$, using the DBSCAN clustering algorithm. Train and test sets were generated from individual sub-domains and their combinations ($_3C_2 = \binom{3}{2} = 3$). Results showed that when the train set and test set included the same sub-domain, the soft domain gap decreased and higher mIoU values were achieved. Conversely, when different sub-domains were employed in the train set and test sets, the soft domain gap emerged and the mIoU values deteriorated.

Table 10: Preliminary study to evaluate soft domain shift between train and test sets. Columns represent train sets, and rows represent test sets. Smaller soft domain gaps lead to the highest performances (mIoU; **Bold**).

| mIoU | D(1) | D(2) | D(3) | D(1, 2) | D(1, 3) | D(2, 3) | D(1, 2, 3) |
|---|---|---|---|---|---|---|---|
| D(1) | **63.93%** | 61.78% | 61.84% | 62.83% | 62.55% | 61.91% | 62.54% |
| D(2) | 61.64% | **63.75%** | 61.63% | 62.20% | 61.52% | 62.73% | 62.78% |
| D(3) | 60.63% | 60.39% | **62.58%** | 60.40% | 61.63% | 61.84% | 61.03% |
| D(1, 2) | 62.01% | 61.87% | 61.75% | **63.94%** | 61.99% | 61.84% | 62.52% |
| D(1, 3) | 62.27% | 62.14% | 62.21% | 62.18% | **64.24%** | 62.27% | 62.28% |
| D(2, 3) | 61.34% | 61.45% | 61.23% | 61.41% | 61.29% | **63.39%** | 62.32% |
| D(1, 2, 3) | 62.69% | 62.49% | 62.46% | 62.68% | 62.57% | 62.60% | **64.67%** |

# D METHODS

## D.1 DESIGN PRINCIPLE OF SIEVE LOSS

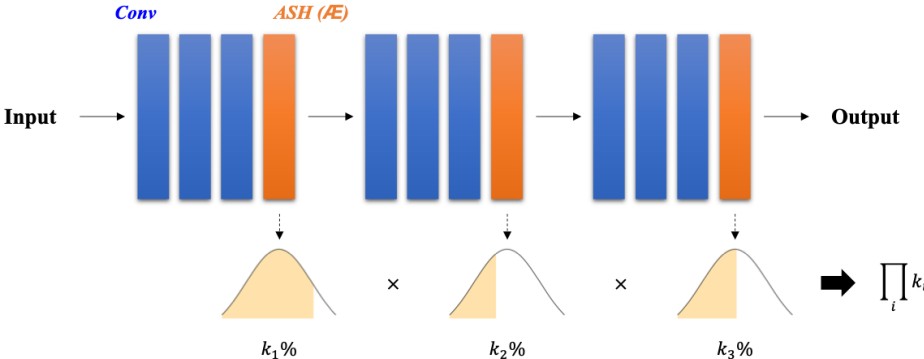

Figure 11: Semantic pipeline of Sieve Loss. Blue blocks are convolution operators, and orange blocks are ASH activation functions.

In CNNs, activation functions such as ReLU rectify the output of convolution operators. Prior research Lee et al. (2022) has, however, suggested a rectified ratio of how many important features are passed. As illustrated in Appendix Figure D.1, the sieve loss is designed based on the ASH activation function. Information and latent features pass through convolution operators and activation functions, and the ASH activation functions rectify the information. Figure 5 can be used to calculate the portion of important features from the input. Since the portion is successively accumulated, the final ratio of the rectification from the input is the product of all rectification levels of ASH activation functions. The ASH activation function provides the rectification levels as attention levels, and the attention level is indicated by the area under the curve of a Gaussian distribution. Therefore, each $k_i$ in Appendix Figure 3 is calculated as follows:

$$k_i = \int_{-\infty}^{i} \frac{1}{\sqrt{2\pi}} e^{-\frac{x^2}{2}} \, dx \tag{17}$$

In density-based classification or segmentation tasks, the important features or information should be related to the area of the target objects in order to accurately predict their density. Sieve loss is designed to reduce the discrepancy between the predicted density by Equation (17) and the labels for the density of the target objects. The sieve loss is defined as follows:

$$\sum_{\mathcal{X},\mathcal{Y} \in \mathbb{X}, \mathbb{Y}} \left\| \prod_{\in \mathcal{A}(\theta_{\mathrm{SDC}})} \left( \int_{-\infty} \frac{1}{\sqrt{2\pi}} e^{-\frac{x^2}{2}} \, dx \right) - d^c(\mathcal{X}|\mathcal{Y}) \right\|_2 . \tag{18}$$

In the training of SDA-Net, the SDC is optimized via the sieve loss for the density-based classification using the calculated density of the labels. During the fine-tuning of SDA-Net, the BN is fine-tuned using the sieve loss for the segmentation task, taking into account the pseudo-predictions and the predicted density index provided by the SDC.

# E  EXPERIMENTS

## E.1  HISTOGRAM OF DENSITY

The density ratio is not the ratio of the number of pixels in the dataset, but rather the proportion of pixels in an individual image relative to the total number of pixels in that image. Bars indicate 95% confidence intervals, and points (fliers) represent the data that extend beyond the whiskers.

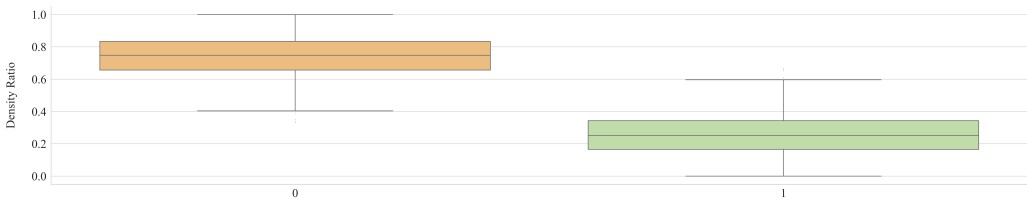

Figure 12: Whiskers plot for density ratios of the WHU dataset

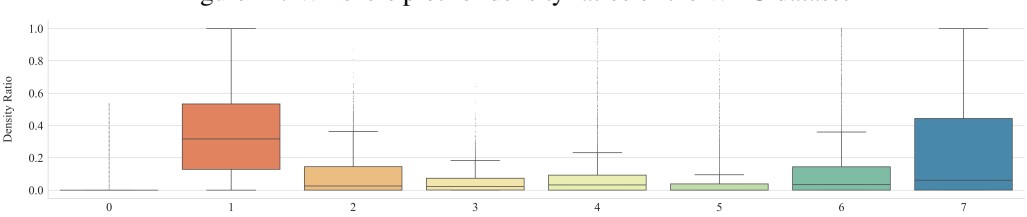

Figure 13: Whiskers plot for density ratios of the LoveDA dataset

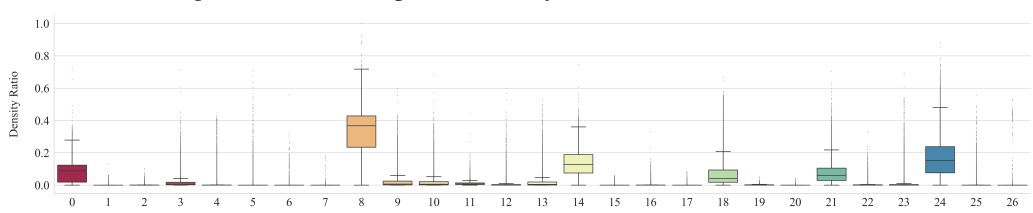

Figure 14: Whiskers plot for density ratios of the GTA5 dataset

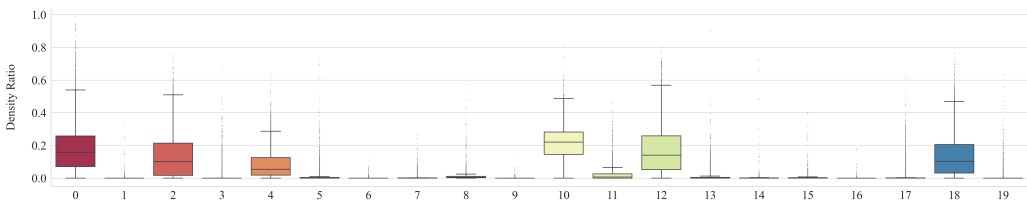

Figure 15: Whiskers plot for density ratios of the BDD100K dataset

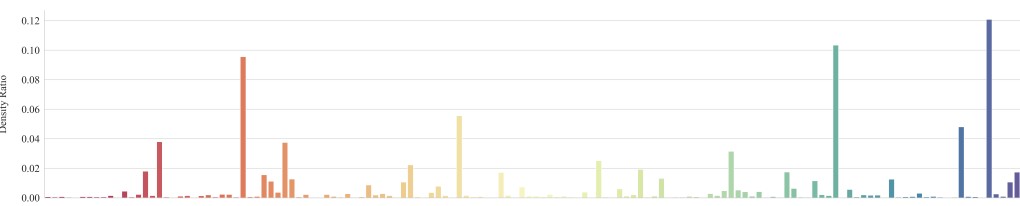

Figure 16: Whiskers plot for density ratios of the ADE20K dataset

## E.2 VIOLIN CHART FOR SDA-NET AND OTHER DL MODELS USING LOVEDA DATASET

Since the sub-domain-wise segmentation brings different accuracy on each sub-domains, the SDA-Net shows an un-uniform distribution in the violin chart, compared with other DL models.

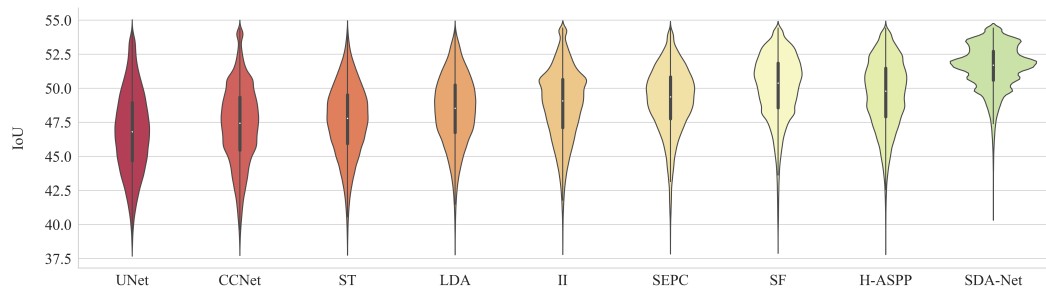

Figure 17: Violin chart for SDA-Net and other DL models.

