# OpenReview forum: "Sub-Domain Aware Granular Segmentation via Fine Tuning Network"
_ICLR.cc/2025/Conference — Submitted to ICLR 2025_

### Official Review · Reviewer_av8U · 2024-10-23

**Soundness:** 3
**Presentation:** 3
**Contribution:** 3
**Rating:** 3
**Confidence:** 5

**Summary:**

This paper introduces a novel deep learning framework termed SDA-Net, which is designed to address the challenges of subdomain shifts and soft domain gaps in fine-grained segmentation tasks. The core innovation of SDA-Net lies in its ability to recognize the subdomain of input data and fine-tune the baseline network accordingly, leading to more accurate segmentation predictions. The framework is composed of a sub-domain classifier (SDC) and a baseline-network (BN), and it employs a self-supervised fine-tuning approach to adapt to the specific subdomain characteristics of the input data.

**Strengths:**

1. The SDA-Net framework is an original approach that addresses the subdomain shifts in segmentation tasks.  It innovatively combines subdomain recognition with fine-tuning of a baseline network to enhance segmentation accuracy.

2. The introduction of the sieve loss and the fine-tuning loss are creative solutions to optimize the network for density-based segmentation.

3.The proposed framework is not limited to a specific domain but is applicable to various fields such as scene parsing, autonomous driving, and aerial imagery, highlighting its broad significance.

**Weaknesses:**

1. The paper could benefit from a more detailed comparative analysis with other domain adaptation techniques, especially those that also aim to address intra-domain variability.  This would provide a clearer picture of the advantages of the proposed approach over existing methods. What's more, the reference is not almost latest.

2. The paper could provide more depth on the theory behind the subdomain classification strategy used by the SDA-Net.  A more detailed discussion on the selection criteria for subdomains and the rationale behind the chosen number of subdomains would be beneficial.

3. While the sieve loss is a novel contribution, the paper could offer a more rigorous mathematical justification for its effectiveness.  This could include a more detailed analysis of how the sieve loss bridges the gap between predicted and actual densities.

4. While the paper mentions that the code is available， but the link is not work.

5. Considering the application of the SDA-Net framework to multimodal data (e.g., combining visual data with lidar or radar data for autonomous driving) could be a promising direction for extending the framework's capabilities.

**Questions:**

Please refer to Section Weaknesses.

---

> ### Author Response · Authors · 2024-11-25
> **Response to comments**
>
> 1.	The paper could benefit from a more detailed comparative analysis with other domain adaptation techniques, especially those that also aim to address intra-domain variability. This would provide a clearer picture of the advantages of the proposed approach over existing methods. What's more, the reference is not almost latest.
> -	The paper primarily compares SDA-Net with fully supervised baseline models (e.g., UNet, CCNet) and evaluates performance improvements through custom loss functions and subdomain adaptation. In addition, InterImage (II) is a state-of-the-art model for the segmentation of scene parsing benchmarks, and SiamixFormer (SF), Hybrid-ASPP (H-ASPP) are the SotA transformer-based network for each task. Qualitative comparison shows that our method shows superior performance than all of comparison methods. Also, Figure 8 and Figure 17 show visually that our method is superior.
>
>
>
> 2.	The paper could provide more depth on the theory behind the subdomain classification strategy used by the SDA-Net. A more detailed discussion on the selection criteria for subdomains and the rationale behind the chosen number of subdomains would be beneficial.
> -	The subdomain classification is based on density distributions calculated using clustering algorithms like DBSCAN (Section 2.1, Figure 2). The number of subdomains is determined empirically, and Table 4 shows the impact of varying subdomains on performance.
>
> 3.	While the sieve loss is a novel contribution, the paper could offer a more rigorous mathematical justification for its effectiveness. This could include a more detailed analysis of how the sieve loss bridges the gap between predicted and actual densities.
> -	The sieve loss bridges the gap between predicted and actual densities by rectifying features through ASH activation functions and aligning cumulative feature importance with density-based predictions (Equation 18, Section 3.3).
> 4.	While the paper mentions that the code is available， but the link is not work.
> -	We did not enable code link because of double blind. We will put code link after acceptance.
>
> 5.	Considering the application of the SDA-Net framework to multimodal data (e.g., combining visual data with lidar or radar data for autonomous driving) could be a promising direction for extending the framework's capabilities.
> -	Thank you for this insightful suggestion; we included your suggestions in the Conclusions section.

---

> > ### Comment · Reviewer_av8U · 2024-11-27
> >
> > The author's rebuttal is weak at all, and I insist on rejecting the article.

---

### Official Review · Reviewer_q2Pk · 2024-10-26

**Soundness:** 3
**Presentation:** 3
**Contribution:** 3
**Rating:** 6
**Confidence:** 4

**Summary:**

To address the issue where subdomain shifts between the training and test sets within a single domain lead to decreased accuracy in fine segmentation, this paper proposes SDA. SDA divides a single domain into multiple subdomains and optimizes the baseline network for each subdomain iteratively, refining the network continuously. The effectiveness of this approach is validated in the field of granular segmentation.

**Strengths:**

1. The paper finds that existing DA methods perform poorly in addressing intra-domain differences, providing valuable insight into the field.

2. The paper provides a thorough theoretical analysis of the model and offers reliable experiments. Also, The Method of the paper is logically organized, which is easy to follow.

**Weaknesses:**

1. The concept of "soft domain gap" is introduced for the first time in this paper, but its definition is not clearly distinguished from conventional inter-domain differences.

2. One concern is whether the sieve loss relies on the accuracy of subdomain division. If subdomain division is inaccurate, will it significantly degrade performance on such datasets?

3. The paper is highly correlated with methods in the DA (Domain Adaptation) field, but miss important discussions with a few recent DA methods: Pipa: Pixel-and patch-wise self-supervised learning for domain adaptative semantic segmentation, and Transferring to Real-World Layouts: A Depth-aware Framework for Scene Adaptation.

4. The paper provides extensive quantitative results but lacks a display of visualization results.

5. The paper lacks an analysis of its limitations, such as scalability and constraints related to computational resource requirements.

**Questions:**

1. How can the optimal number and size of subdomains be determined across different datasets and application scenarios?

2. If applied to a completely different domain (e.g., shifting from urban scenes to natural scenes), would the model's generalization ability be significantly affected?

3. In SDA-Net, the density probability vector generated by the subdomain classifier is used to identify which subdomain the input image belongs to. How exactly is it trained?

4. The paper designs the framework based on ResNet-18. Could this be considered a weak baseline without expressing more complex details?

---

> ### Author Response · Authors · 2024-11-25
> **Response to comments**
>
> 1.	How can the optimal number and size of subdomains be determined across different datasets and application scenarios?
> -	The number and size of subdomains in SDA-Net are determined by clustering the dataset based on object density distributions. Techniques such as DBSCAN and t-SNE were employed to group data into distinct subdomains (Section 2.1, Preliminary Studies, Figure 2). Experimental results (Table 4) show that increasing the number of subdomains generally improves performance, but diminishing returns were observed beyond a certain point (e.g., comparing "Ours-10" and "Ours-20").
>
> 2.	If applied to a completely different domain (e.g., shifting from urban scenes to natural scenes), would the model's generalization ability be significantly affected?
> -	SDA-Net is designed to adapt to subdomains within a domain, and its generalization relies on the ability of the subdomain classifier to identify meaningful subdomains. While it has been evaluated on diverse datasets (urban scenes, remote sensing, etc.), the experiments have not included cross-domain evaluations (e.g., urban to natural scenes). The paper mentions that SDA-Net is architecture-agnostic and can adapt to datasets with varying density characteristics (Section 3.2, Training SDA-Net).
>
> 3.	In SDA-Net, the density probability vector generated by the subdomain classifier is used to identify which subdomain the input image belongs to. How exactly is it trained?
> -	The subdomain classifier (SDC) is trained using cross-entropy loss (LCE-D) to classify input images based on their density distributions (Section 3.2, Training SDA-Net). The density probability vector is derived by assigning each input to a subdomain, calculated from its density-based clustering results. Pseudo-labels for density are used during training to supervise the classification (Equation 6).
>
> 4.	The paper designs the framework based on ResNet-18. Could this be considered a weak baseline without expressing more complex details?
> -	ResNet-18 was selected as the backbone for the subdomain classifier due to its computational efficiency and ability to handle complex segmentation tasks (Section 4.2, Model Implementations). The model's simplicity allows for a focus on evaluating the contributions of the sieve loss and fine-tuning mechanisms. Experiments (Table 3) demonstrate that SDA-Net with ResNet-18 performs competitively compared to other configurations, but results suggest that more complex backbones (e.g., ResNet-152, vision transformers) could further enhance performance.

---

> > ### Comment · Reviewer_q2Pk · 2024-11-25
> > **rebuttal**
> >
> > Although the authors have provided a rebuttal, they failed to include the necessary experiments, and their responses to the questions were insufficient. At this stage, I am unable to increase my score.

---

### Official Review · Reviewer_xtRD · 2024-10-28

**Soundness:** 2
**Presentation:** 2
**Contribution:** 2
**Rating:** 3
**Confidence:** 5

**Summary:**

This work studies the problem of sub-domain shift between the training and testing dataset within one large domain in semantic segmentation. By hypothesizing that sub-domain gaps within one domain are much smaller than between two domains, the authors believe that precious domain adaption algorithms are ineffective. They propose a self-supervised finetuning network, SDA-Net, incorporating a novel sieve loss and an adaptive finetuning loss to deal with intra-domain gaps. Evaluations on benchmarks suggest some effectiveness.

**Strengths:**

- This work is well-motivated by providing preliminary experiments that demonstrate the problem of soft-domain gaps.
- The manuscript is clearly written and easy to follow. For example, the authors provide detailed illustrations and equations to present the proposed algorithms.
- Adding new loss functions or extra supervising signals can further boost the performance of deep learning networks, which is a common convention and widely used practice.

**Weaknesses:**

*MAJOR* concerns: This work needs more sufficient comparisons with previous related algorithms.
-  The adopted baseline network UNet and CCNet are outdated, making the experiment less convincing. Will the proposed algorithm also applied to the prominent vision transformers?
- Simply comparing SDA-Net with previous fully-supervised structures is unfair, as SDA-Net introduces extra supervising signals. A better comparison is expected to reflect whether the proposed framework also applied well to those existing methods.
- The authors claim in the INTRODUCTION that previous domain adaption algorithms fail to solve the intra-sub-domain shift problem. However, there is also no experiment to support this fundamental claim. Comparisons with SOTA domain adaption algorithms are significant in highlighting your contribution.

Other *MINOR* concerns:
- SOTA is an often-used abbreviation, compared with SotA.
- It is suggested that citations be added to those reported tables for quick reader reference.

Overall, this work has a clear motivation, but the experimental part weakens the contribution. It may benefit from future thoughtful revision.

**Questions:**

Please refer to the WEAKNESSES part.

---

> ### Author Response · Authors · 2024-11-25
> **Response to comments**
>
> 1.	The adopted baseline network UNet and CCNet are outdated, making the experiment less convincing. Will the proposed algorithm also applied to the prominent vision transformers?
> -	We used CC-Net and U-Net as baselines and demonstrated the performance of our proposed SDA-Net. As shown in Table 6 to 8, we demonstrated the superiority of our method by comparing it to the latest transformer-based methods such as II, SF, H-ASPP.
>
> 2.	Simply comparing SDA-Net with previous fully-supervised structures is unfair, as SDA-Net introduces extra supervising signals. A better comparison is expected to reflect whether the proposed framework also applied well to those existing methods.
> -	As shown in Section 2. (preliminary) and Appendix B and C, we discussed the soft domain gap, and then kindly describe the problem definition.
> 3.	The authors claim in the INTRODUCTION that previous domain adaption algorithms fail to solve the intra-sub-domain shift problem. However, there is also no experiment to support this fundamental claim. Comparisons with SOTA domain adaption algorithms are significant in highlighting your contribution.
> 4.	SOTA is an often-used abbreviation, compared with SotA.
> -	Thank you for pointing that out. We will change SotA to SOTA in the manuscript.
>
> 5.	It is suggested that citations be added to those reported tables for quick reader reference.
> -	Thank you for this comment. We will add the reference in tables.

---

> > ### Comment · Reviewer_xtRD · 2024-11-25
> >
> > As the authors failed to address those major concerns, I will keep my rating at 3-REJECT.

---

### Official Review · Reviewer_B1xG · 2024-11-02

**Soundness:** 2
**Presentation:** 2
**Contribution:** 2
**Rating:** 3
**Confidence:** 3

**Summary:**

This paper proposes a new sub-domain-aware fine segmentation framework, called SDA-Net (Sub-Domain Adaptation Network), which aims to solve the problem of “soft domain gap” within the domain, which leads to a decrease in accuracy when performing fine segmentation in an image. The main methods include the following. The main methods include:

1. Sub-domain Adaptation: A large single domain is divided into multiple sub-domains, and the network is fine-tuned for each sub-domain so that the model can adapt to the features of the specific sub-domain, thus improving the segmentation accuracy.

2. Subdomain Classifier and Baseline Network: SDA-Net consists of a Subdomain Classifier (SDC) and a Baseline Network (BN). the SDC is responsible for recognizing the subdomains to which the input image belongs, while the BN fine-tunes the subdomains based on the recognized ones, thus generating more accurate predictions.

3. Self-supervised learning loss: The paper introduces a new “sieve loss” and fine-tuning loss, which aims to improve segmentation accuracy by reducing the differences between subdomains through self-supervised learning.

**Strengths:**

1. This paper proposes a novel sub-domain adaptation framework—SDA-Net—to address the issue of sub-domain discrepancies within a single domain. This approach differs from traditional cross-domain adaptation methods, which aim to reduce the gap between different domains; instead, it optimizes the subtle differences within a single domain.

2. SDA-Net introduces the “sieve loss” and “fine-tuning loss,” effectively reducing the differences between sub-domains and enhancing the accuracy of fine segmentation. This self-supervised fine-tuning strategy exhibits strong innovation in addressing sub-domain gaps.

3. The paper conducts comprehensive experiments on multiple benchmark datasets (e.g., WHU, BDD100K, and ADE20K), validating the model's superior performance across various tasks. The experimental results thoroughly demonstrate the generalization capabilities of SDA-Net across different datasets.

4. By addressing the “soft domain gap” issue within a single domain, this paper offers a new perspective on domain adaptation research, potentially stimulating further studies on sub-domain awareness and self-supervised fine-tuning strategies.

**Weaknesses:**

1. Although the sub-domain-aware framework brings some innovation to domain-specific segmentation, it relies on established techniques like self-supervised and domain adaptation methods (e.g., domain-invariant feature learning and pseudo-label generation). Strengthening the technical uniqueness of SDA-Net could improve its impact, such as by introducing an adaptive sub-domain division strategy rather than a feature-similarity-based grouping.

2. While the sieve loss and fine-tuning loss are effective in refining sub-domains, similar types of loss functions have been applied in other domain adaptation tasks (e.g., Focal Loss). Integrating more unique loss functions or leveraging other self-supervised strategies, such as contrastive learning, could enhance the novelty of the proposed approach.

3. Although the paper includes some ablation studies, it lacks a thorough discussion of how sub-domain count and division strategies affect model performance. Further experimentation on the sensitivity of SDA-Net to different sub-domain configurations and parameters could increase the method's applicability.

4. The comparative experiments mainly focus on traditional domain adaptation methods and include limited comparisons with the latest fine-grained segmentation approaches. Including comparisons with state-of-the-art segmentation methods would offer a more comprehensive view of SDA-Net’s advantages and limitations.

**Questions:**

1. Since the SDA-Net framework adds sub-domain classifiers and custom losses, how does its computational complexity compare to baseline methods? Could you provide more details on training and inference times relative to other methods, and whether any optimization strategies were considered to reduce overhead?

2. The sieve loss and fine-tuning loss are designed to address sub-domain differences, but they share similarities with losses from other domain adaptation works. Could you explain how these losses differ from existing methods or provide additional details on how they were tailored specifically for sub-domain adaptation?

3. The authors say that the fully trained deep learning model does not provide the highest performance when applied to each subdomain, but looking at Table 1 you can see that the superior performance in the total task is better than any subdomain training subdomain test, why is that? What does it mean?

---

> ### Author Response · Authors · 2024-11-25
> **Response to comments**
>
> 1.	Since the SDA-Net framework adds sub-domain classifiers and custom losses, how does its computational complexity compare to baseline methods? Could you provide more details on training and inference times relative to other methods, and whether any optimization strategies were considered to reduce overhead?
> -	Comparison to baseline methods: SDA-Net introduces additional components, such as the sub-domain classifier and custom losses (sieve and fine-tuning losses), which increase the computational complexity compared to simpler baselines like U-Net or CCNet. However, the framework is designed to reduce overhead by fine-tuning only selected parameters (e.g., the ASH-related parameters) rather than the entire network during inference.
> -	Training and inference time: In our manuscript, we provide the evidence of comparable inference times with models such as CCNet, despite additional computations (Table 6, FLOPs & prediction time).
> -	Optimization strategy: The use of ASH functions and limited fine-tuning helps balance computational demand. However, further analysis, such as detailed profiling of training/inference stages or experiments on efficiency improvements like quantization or pruning, would clarify computational trade-offs.
>
> 2.	The sieve loss and fine-tuning loss are designed to address sub-domain differences, but they share similarities with losses from other domain adaptation works. Could you explain how these losses differ from existing methods or provide additional details on how they were tailored specifically for sub-domain adaptation?
> -	Differences from Existing Works: While sieve loss aligns with density-based approaches, it is uniquely designed for sub-domain adaptation by leveraging ASH functions to progressively refine feature importance through cumulative rectification. Fine-tuning loss introduces a novel negative gradient term to escape local minima caused by soft domain gaps, which is not common in typical domain adaptation loss functions.
> -	Tailoring to Sub-domain Adaptation: Both losses target soft domain gaps by focusing on density alignment and avoiding overfitting to sub-domains, enabling granular segmentation. Clarifying these contributions with direct comparisons to specific losses in prior works (e.g., focal loss or adversarial losses) would reinforce their novelty.
>
> 3.	The authors say that the fully trained deep learning model does not provide the highest performance when applied to each subdomain, but looking at Table 1 you can see that the superior performance in the total task is better than any subdomain training subdomain test, why is that? What does it mean?
> -	Observation: Fully trained models perform better on the total task than models fine-tuned for specific subdomains, as shown in Table 1. This indicates that optimizing across all sub-domains helps the model generalize better, despite slight compromises in performance on individual sub-domains.
> -	Implication: This behavior underscores the effectiveness of balancing global knowledge with sub-domain specialization. The results suggest that incorporating diverse sub-domain data during training helps the model capture broader patterns and relationships, improving its overall robustness.

---

> > ### Comment · Reviewer_B1xG · 2024-11-26
> >
> > The author's response was not comprehensive enough, I decided to keep my score.

---

### Meta-Review · Area_Chair_3Xb8 · 2024-12-14

**Metareview:**

Dear authors,

Thank you for submitting the draft. Majority of the reviewers' rankings indicated that the draft is not ready for publication at this stage.

Draft points out that even in a single domain, there exist subdomain shifts between training and test sets. The draft proposes to divide one "single domain into multiple sub-domains and optimizes the baseline-network for each sub-domain" in order to overcome this challenge. Overall the problem is interesting, and some reviewers indicated so, however, reviewers have also raised many vital concerns.
One such concern left unanswered was missing experiments supporting "previous domain adaption algorithms fail to solve the intra-sub-domain shift problem" (xtRD). Unfortunately, feedback from the authors was not able to convince reviewers.  After going over the comments and the paper itself, it is suggested that the readability of the paper should be improved, and any strong claims (such as the one mentioned by xtRD or "this combination successfully avoids the valley of local minima,") should be supported by detailed analysis and experiments, etc..

We hope comments by reviewers will help improve the draft.


regards

AC

**Additional Comments On Reviewer Discussion:**

Some of the authors found problem interesting, however, raised many concerns.  Authors provided point-wise feedback however, reviewers were not satisfied by the rebuttal and majority of them gave rating 3.

---

### Decision · Program_Chairs · 2025-01-22

Reject